# Adsorption of Indium(III) Ions from an Acidic Solution by Using UiO-66

Wanyi Zeng [1,2], Lei Xu [3], Qiongling Wang [4], Chen Chen [3] and Minglai Fu [1,3,*]

1. Key Laboratory of Urban Pollutant Conversion, Institute of Urban Environment, Chinese Academy of Sciences, Xiamen 361021, China; wyzeng@iue.ac.cn
2. University of Chinese Academy of Sciences, Beijing 100049, China
3. Xiamen Key Laboratory of Municipal and Industrial Solid Waste Utilization and Pollution Control, College of Civil Engineering, Huaqiao University, Xiamen 361021, China; lxu@hqu.edu.cn (L.X.); cchen@hqu.edu.cn (C.C.)
4. Quanzhou Quanzhong Vocational Secondary School, Quanzhou 362005, China; zhl@lmu.edu.cn
* Correspondence: mlfu@iue.ac.cn; Tel.: +86-592-6190762

**Abstract:** Considering environmental friendliness and economic factors, the separation and extraction of indium under acidic conditions are of great significance. In this research, metal-organic frameworks (MOFs) of UiO-66 were successfully prepared and used for the separation and adsorption of indium. The properties of UiO-66 were structurally characterized using powder X-ray diffraction (XRD), Fourier-Transform Infrared Spectroscopy (FTIR), Brunauer-Emmett-Teller surface area analyzer (BET), thermogravimetric analysers (TGA) and Scanning Electron Microscope (SEM). The results show that UiO-66 can resist acid and keep its structure unchanged, even at a strong acidity of pH 1. The adsorption performance of UiO-66 to indium (III) was also evaluated. The results show that the adsorption process of indium ions was by the Langmuir adsorption isotherm, with a maximum adsorption capacity of 11.75 mg·g$^{-1}$ being recorded. The adsorption kinetics experiment preferably fits the second-order kinetic model. A possible mechanism for the adsorption of In(III) by UiO-66 was explored through X-ray photoelectron spectroscopy (XPS) and Fourier transform infrared analysis(FT-IR). It was concluded that the C=O of free –COOH of UiO-66 was involved in the adsorption of In(III) by cation exchange. This study indicates, for the first time, that UiO-66 can be applied as an acid-resistant adsorbent to recover indium (III).

**Keywords:** UiO-66; MOFs; Adsorption; Indium recovery

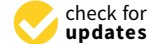



## 1. Introduction

Indium (In), one of the most scarce and critical metal resources of the past few decades, has been greatly exploited in large quantities because of its optical transparency and semiconducting properties [1]. For a long time, 70% of indium around the world has been used for the production of indium tin oxide(ITO) [2]. It is estimated that indium reserves will be depleted in several years based on the calculation of its consumption rate and the available amounts in the Earth's crust. Ironically, electronic products become outdated very quickly, which inevitably results in the accumulation of used electronic waste (e-waste) containing indium resources. It is also notable that indium is a very toxic metal and can be released from e-waste to the environment, causing lung disease and cancer. In this context, the recovery of indium from secondary resources is vital for sustainable development, which is attracting more and more attention [3,4]. More than half of indium resources have been recovered from secondary sources [2,5]. The content of indium in liquid crystal displays (LCDs) is 1400 mg·kg$^{-1}$, while the content of indium in ordinary zinc ore is 10 to 20 mg·kg$^{-1}$. Therefore, LCDs are a secondary resource capable of providing significant amounts of recycled indium [6]. At present, different methods have been reviewed for the adsorption and separation of indium, such as solvent extraction, cation exchange

resin [7], supercritical extraction [8], electrochemical techniques [9], co-precipitation [10], and solid-phase extraction [11].

Extraction is considered one of the most economic methods for the separation of indium due to its various advantages, such as its excellent selectivity, high separation efficiency, and low energy consumption. Tributyl phosphate ($C_{12}H_{27}O_4P$), Cyanex 272 ($C_{16}H_{35}O_2P$), and Cyanex 923 ($C_{24}H_{51}OP$) have been reported as efficient extracting solvents for recovering indium [12]. However, these extractants are slightly soluble in water, and the extraction process will cause part of the solvent to be lost, causing wastewater pollution and economic loss [13]. Meanwhile, the main disadvantage of cation exchange resin is that its adsorption and desorption speed is slow, and electrolyte desorption is required [14].

An advantage of supercritical carbon dioxide extraction is that it is safe and clean, but it needs to be performed at a high pressure (above 20 MPa) [6]. The adsorption process, due to its low cost, simple operation, and high efficiency has become the most popular method for the recovery of indium from wastewater. Clague et al. [15] applied chitosan-coated bentonite (CCB) to adsorb indium ions. Zhang et al. [16] developed $TiO_2$ nanomaterials to separate and adsorb indiums with the greatest adsorption capacity of 3.9–6.5 $g \cdot g^{-1}$ in the temperature range of 0–50 °C. Alguacil et al. [17] applied multiwall carbon nanotubes to separate and adsorb indium ions. Fang et al. [18] applied $CoFe_2O_4$-zeolite materials to separate and adsorb indium ions, recording the highest adsorption capacity of 94.78 $\mu g \cdot g^{-1}$. Mohammad et al. utilized macroporous resins, called TP 260 (resins functionalized with (aminomethyl) phosphonic acid), TP 208 (resins functionalized with iminodiacetic acid), and IRA743 (resins functionalized with N-methylglucamine (free base form)), for the recovery of indium [19].

However, the recovery of indium in acidic solution by metal-organic frameworks (MOFs) has never been studied, despite this being a promising method as recovered indium is usually found in the acidic aqueous solutions produced by e-waste [20,21]. Compared with traditional porous adsorbents, MOFs have the advantages of a simple synthesis, beautiful shape, and size, large specific surface area, strong metal surface activity, adjustability, good crystallization, ordered porous structure, high removal rate, good thermal stability, strong adsorption capacity, strong regeneration, high selectivity, and good reproducibility [22–25].

To date, MOFs have been applied in catalysis [26], drug delivery, separations, sensors, etc. [27,28]. Compared with the precipitation method, MOF materials can be used as an adsorbent for the adsorption of heavy metals at a low concentration, which makes up for the defect that metals have a certain solubility in water and cannot be precipitated for recovery. MOFs have been widely used for heavy metal recovery due to their ease of operation (at room temperature) [29], but they do have several disadvantages, including their instability in acid-base conditions, which has greatly limited their application [30].

In this paper, the adsorption of In(III) on an acid-resistant MOF material of UiO-66 was used to absorb and recover indium in a solution for the first time. UiO-66 is one of the MOFs that contains the $[Zr_6(\mu_3\text{-}O)_4(\mu_3\text{-}OH)_4 (COO)_{12}]$ cluster with a linker length [27]. The structure of UiO-66 was characterized by powder X-ray diffraction (XRD), Fourier-Transform Infrared Spectroscopy (FTIR), Brunauer-Emmett-Teller surface area analyzer (BET), thermogravimetric analysers (TGA) and Scanning Electron Microscope (SEM). The adsorption performance of UiO-66 to In(III) was evaluated. The effects of reaction time, coexisting ions, pH value, and other factors affecting the adsorption capacity were studied. The mechanism of indium adsorption and desorption of UIO-66 in acid solution was described. The highlight of this study was the development of acid-resistant MOFs for the recovery of In(III).

## 2. Materials and Methods

### 2.1. Chemicals and Materials

$ZrCl_4$, hydrochloric acid (HCl), N-N-dimethylformamide (DMF), anhydrous ethanol, $AlCl_3 \cdot 6H_2O$, and $MgCl_2 \cdot 6H_2O$ were obtained from Sinopharm Chemical Reagent Co., Shanghai, China. $H_2BDC$ (1, 4-benzene-dicarboxylic acid) and $InCl_3 \cdot 6H_2O$ were obtained from Aladdin Industrial Corporation, Shanghai, China. The purchased experimental reagents were analytically pure and were able to be used without purification. The standard solution of indium (1000 mg·$L^{-1}$) was provided by the National Research Center of China, Beijing, China. The $In^{3+}$ stock solution was prepared by dissolving $InCl_3 \cdot 6H_2O$ in HCl solution.

### 2.2. Preparation of UiO-66

The preparation of UiO-66 was done according to our previously reported method [31]. In a typical synthesis, $ZrCl_4$ (6.4 mmol) and $H_2BDC$ (6.4 mmol) were dissolved together in 120 mL of DMF (DMF=N, N-dimethylformamide) under stirring, and transferred to a 200 mL polytetrafluoroethylene reactor. The autoclave was kept at 80 °C and then transferred to 120 °C. The white powder was washed with ethanol and water, centrifuged, and then dried overnight at a drying temperature of 80 °C.

### 2.3. Material Characterization

The crystal structure was analyzed by XRD (Rigaku-TTRIII, PANalytical corporation, Almelo, Holland) with Cu Kα radiation, a scan range from 10° to 80° at 40 kV voltage, and a 40 mA current. The functional groups of samples were identified by employing Fourier transform infrared spectroscopy (Thermo Scientific Nicolet iS10, Thermo Fisher Scientific, Waltham, MA, USA). The pore volume was analyzed by the $N_2$ adsorption and desorption technique using Brunner–Emmett–Teller theory (BET) (ASAP 2020 Micro metrics, Micromeritics, Norcross, GA, USA). Thermal gravimetric analysis (TGA) was conducted on NETZSCH TG 209 F3 Instruments (NETZSCH Scientific Instruments Co., Selb, Germany) in air atmosphere with the temperature range of 40–800 °C at 10.0 °C/min heating rate. The surface morphology was analyzed by scanning electron microscopy (SEM, HITACHI S-4800, Hitachi, Japan) and indium concentration was measured using inductively coupled plasma optical emission spectrometry (ICP-OES) (Optima 7000 DV PerkinElmer, Waltham, MA, USA). Zr concentration was measured using ICP-OES (Agilent 5110, Palo Alto, CA, USA) and X-ray photoelectron spectroscopy (XPS) (Kratos Amicus, Shimadzu, Japan).

### 2.4. Adsorption Isotherm Experiments

Different concentrations of indium solutions (pH = 3) were configured for adsorption isotherm experiments. The concentration of indium was configured to 10–80 mg·$L^{-1}$. A total of 10 mg of MOFs was added into the indium solution of different concentrations in Erlenmeyer flasks. The Erlenmeyer flasks were shaken for 24 h, at 25 °C and 200 revolutions per minute on a shaker. For the adsorption kinetics curve, 200 mg of MOFs was placed in 400 mL of indium solution (pH = 3.0 ± 0.1, $C_0$ = 20 mg·$L^{-1}$), and a solution sample of 2 mL was taken at regular intervals. The hydrochloric acid solution was used to adjust the pH of the indium solution. To evaluate the interference of coexisting cations, magnesium chloride or aluminum chloride was added into the $In^{3+}$ solution. After sampling, the samples were filtered with a 0.22 μm membrane. All experiments were conducted in 2 parallel experiments to get the experimental results.

The regeneration experiment was conducted by separating UiO-66 from an $In^{3+}$ solution of pH 3 by centrifugation, and then this was soaked in an HCl solution of pH 1 for desorption.

The amount of adsorbed $In^{3+}$ (Equation (1)), the first-order kinetic model (Equation (2)), the second-order kinetic model (Equation (3)), the Langmuir isotherm model (Equation (4)), and the Freundlich isotherm model (Equation (5)) were determined as follows:

$$q_t = \frac{(c_0 - c_t) \times V}{m}, \tag{1}$$

$$\ln(q_e - q_t) = \ln q_e - k_1 t, \tag{2}$$

$$\frac{t}{q_t} = \frac{1}{k_2 q_e^2} + \frac{t}{q_e}, \tag{3}$$

$$\frac{C_e}{q_e} = \frac{1}{K_L q_{max}} + \frac{C_e}{q_e}, \tag{4}$$

$$\ln q_e = \ln K_f + \frac{1}{n} \ln C_e, \tag{5}$$

where $q_t$ (mg·g$^{-1}$) represents the adsorption amount of $In^{3+}$ sampled, and $c_0$ (mg·g$^{-1}$) represents (mg·g$^{-1}$) represent the initial concentration and sampling concentration of indium, respectively. $m$ (g) presents the mass of the adsorbent, while $V$ (L) represents the volume of the $In^{3+}$ solution. Ce corresponds to the concentration of indium in equilibrium (mg·g$^{-1}$), and $q_e$ and $q_{max}$ represent the adsorption capacity of the material (mg·g$^{-1}$) at the equilibrium time and maximum time, respectively. $k_1$ (min$^{-1}$) and $k_2$ (g·mg$^{-1}$·min$^{-1}$) correspond to the rate constants of the pseudo-first-order and second-order adsorption models, respectively. $K_L$ is the Langmuir constant of the adsorption energy (L·g$^{-1}$). $K_f$ corresponds to adsorption capacity and $1/n$ corresponds to the adsorption intensity.

## 3. Results

### 3.1. Characterization of Discussion

The crystal structures of as-synthesized UiO-66 and UiO-66 after soaking in an acid solution of pH 1 and 2 were characterized by XRD. As shown in Figure 1a, the characteristic peaks of UiO-66 were located at 7–9°, which were consistent with previous literature reports [32,33]. The result indicated the successful formation of UiO-66 MOFs. When UiO-66 was soaked in the acid solution of pH 1 and 2 for 12 h, it was found that the peak positions of UiO-66 remained unchanged, indicating that UiO-66 was stable and maintained its original structure under acute acidic conditions. The result suggests that UiO-66 could be possibly applied to the separation of $In^{3+}$ in a solution without the requirement of adjusting the pH after leaching $In^{3+}$ from LCDs by soaking it with an acidic medium.

FT-IR spectra were used to characterize UiO-66 and UiO-66 after soaking in an acid solution of pH 1 and 2 for 12 h. As shown in Figure 1b, for UiO-66, the characteristic peak of Zr-O of the UiO-66 peaks was located at 744 cm$^{-1}$, and the vibration peaks of the aromatic benzene rings were located at 1402 cm$^{-1}$, 1570 cm$^{-1}$, and 1654 cm$^{-1}$ [31]. This indicates that UiO-66 was successfully synthesized. Furthermore, the spectra of UiO-66 and UiO-66 after soaking in acid solution were almost the same, which also indicates the stability of UiO-66 in the acid condition.

As shown in Figure 1c, the adsorption isotherm of the nitrogen adsorption-desorption was studied. The adsorption and desorption isotherm of UiO-66 had micropores under a very low relative pressure, meaning it was a typical type I [34,35]. The BET specific surface area was 749.36 m$^2$·g$^{-1}$, showing the high surface area of the UiO-66 material. Furthermore, UiO-66 recorded a pore volume of 0.67 cm$^3$/g.

TGA can be used to evaluate the thermal stability of UiO-66. The experiment was carried out in air at 40–800 °C. As shown in Figure 1d, below 200 °C, the weight loss was 11.5%, which may have been due to the water molecules that were adsorbed to the surface of UiO-66 and that evaporated below 200 °C. At 200–400 °C, the weight loss was 6.7%, which may have been due to the evaporation of residual ligands. At 400–550 °C, the weight loss was 35.5%, which may have been due to the decomposition of UiO-66 into $ZrO_2$. The

TGA profiles suggested that the prepared MOFs have excellent thermal stability, which is similar to the Zr-MOFs reported previously [31].

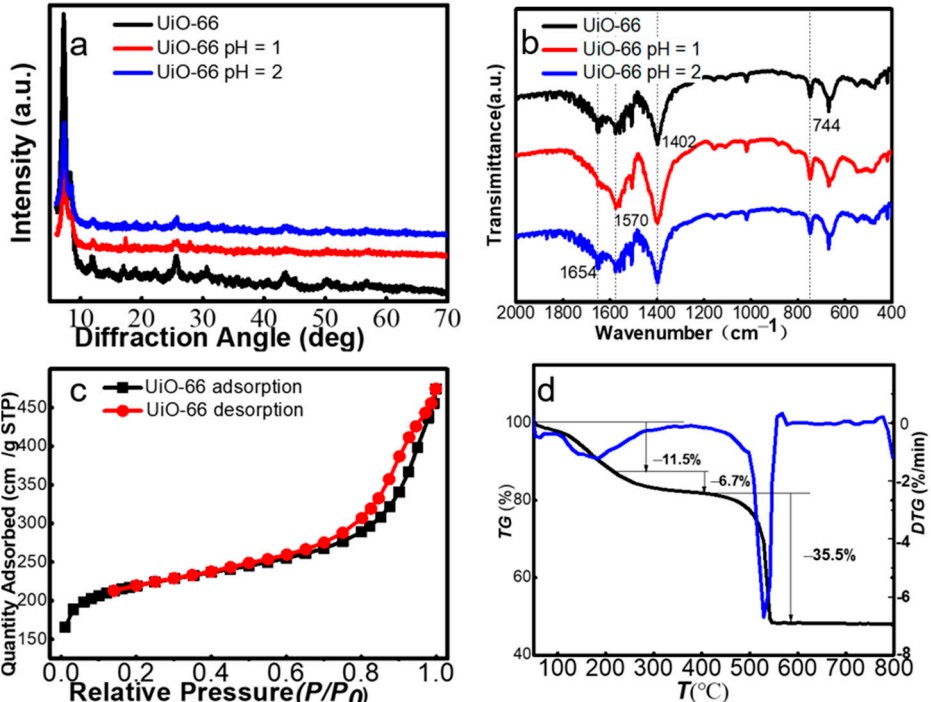

**Figure 1.** (**a**) XRD patterns of UiO-66 and UiO-66 after soaking in an acid solution of pH 1 and 2 for 12 h; (**b**) FT-IR spectra of UiO-66 and UiO-66 after soaking in an acid solution of pH 1 and 2 for 12 h; (**c**) $N_2$ adsorption-desorption isotherms of UiO-66; (**d**) TG and DTG curves of UiO-66.

Determining the Zr concentration in the acidic solutions after contacting them with UiO-66 is important for further supporting the acidic resistance of UiO-66. Therefore, determination of the concentration of Zr of UiO-66 in 0.1M HCl solution was conducted. If zirconium was completely soluble in acid, the concentration of our solution was 100 ppm. However, the Zr concentration obtained by the ICP-OES test was 0.17 ppm; considering the test error, this concentration was close to 0, indicating that UiO-66 was not dissolved in the acid solution. This conclusion was consistent with the results obtained by XRD and FT-IR.

SEM images of UiO-66 and UiO-66 after soaking in an acid solution of pH 1 for 12 h are provided in Figure 2. As shown in Figure 2a–c, the size of the UiO-66 nanoparticles ranged from 20 to 40 nm. When UiO-66 was soaked in the acid solution of pH 1 for 12 h, it was found that the morphology of UiO-66 remained unchanged (Figure 2c), indicating that UiO-66 was stable and maintained its original structure under acute acidic conditions. This result was consistent with those indicated by XRD and FT-IR. Therefore, all of these observations demonstrate that the formation of UiO-66-MOFs that can resist acid and keep their structures intact (even at a strong acidity of pH 1 it was successful).

### 3.2. Adsorption Experiments

#### 3.2.1. Adsorption Kinetic

To simulate the adsorption kinetic curve of indium ions, UiO-66 was put into the indium solution to adsorb indium, and the concentration of the solution sample was tested at intervals [31]. The obtained results are shown in Figure 3 and Table 1. The adsorption kinetics of the UiO-66 adsorption of indium conformed to the second-order kinetic curve with the correlation coefficient ($R^2$) of 0.960. In the first 6 h, the adsorption was relatively fast, and then the adsorption rate slowly decreases until the equilibrium was reached in 24 h.

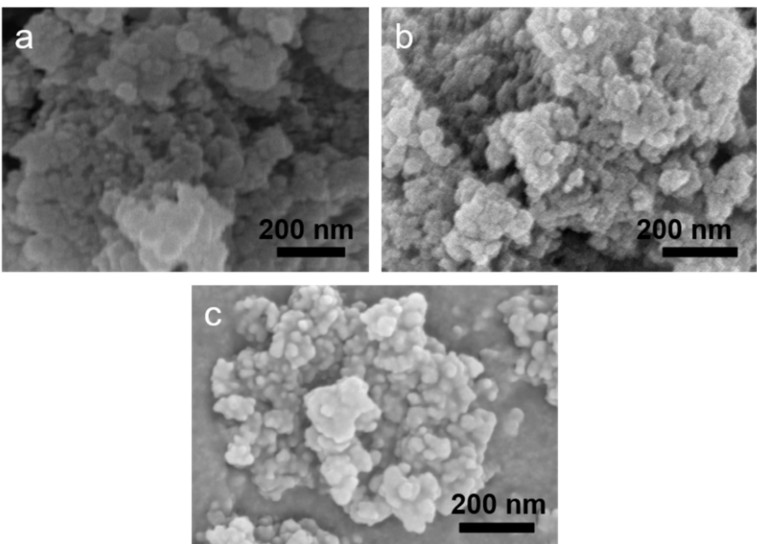

**Figure 2.** SEM of UiO-66 (**a**), UiO-66 after soaking in acid solution of pH = 1 for 12 h (**b**) and UiO-66 after soaking in acid solution of pH = 2 for 12 h (**c**).

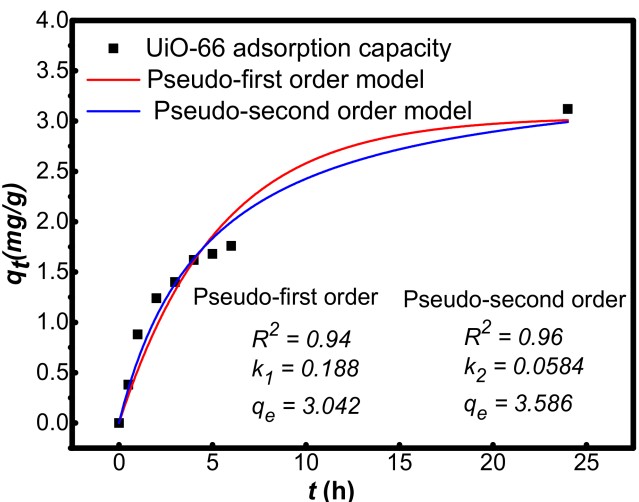

**Figure 3.** Kinetic curve of indium adsorption of UiO-66 (pH = 3, *m* (UiO-66) = 200 mg, *V* (solution) = 400 mL, $C_0$ = 21.3 ppm and *T* = 298.15 K).

**Table 1.** Adsorption kinetic model parameters of UiO-66.

| Adsorbent | Pseudo-First Order Model | | | Pseudo-Second Order Model | | |
|---|---|---|---|---|---|---|
| | $k_1$ (min$^{-1}$) | $q_e$ (mg·g$^{-1}$) | $R^2$ | $k_2$ (g·mg$^{-1}$·min$^{-1}$) | $q_e$ (mg·g$^{-1}$) | $R^2$ |
| UiO-66 | 0.188 | 3.042 | 0.941 | 0.0584 | 3.586 | 0.961 |

### 3.2.2. Adsorption Isotherm Tests

Figure 4 and Table 2 show the adsorption isotherms of In(III) onto UiO-66 at 298 K. The correlation coefficient ($R^2$) of the Freundlich model was 0.987, while the correlation coefficient ($R^2$) of the Langmuir model was 0.996. The experimental data were more accurately fitted by the Langmuir model. Table 1 lists the adsorption saturation capacities of different indium adsorbents under the corresponding experimental conditions reported in the literature. The adsorption capacity of UiO-66 indium was 11.75 mg·g$^{-1}$. The statistical data in Table 3 show that UiO-66 recorded a higher indium adsorption capacity than CCB, $TiO_2$, and $CoFe_2O_4$-zeolite composite [15,16,18,36]. while it recorded a lower

indium adsorption capacity than TP207 [36], poly(vinyl phosphonic acid-co-methacrylic acid) [37], ambulate IRA-400AR [38], modified solvent impregnated resins (MSIRs) [38], Amberlite IRA-400AR [39], and coated solvent impregnated resins [13]. The adsorption saturation of indium by the adsorbent was affected by many factors, such as the pH of the solution, the amount of adsorbent used, temperature, and the initial concentration of indium. A higher adsorbent dose meant that there were more binding sites available to absorb indium (III) onto the surface of the adsorbent. A higher initial concentration meant there was a better concentration gradient, which is an important driving force to overcome the mass transfer resistance of indium (III) between the liquid and solid phases [15].

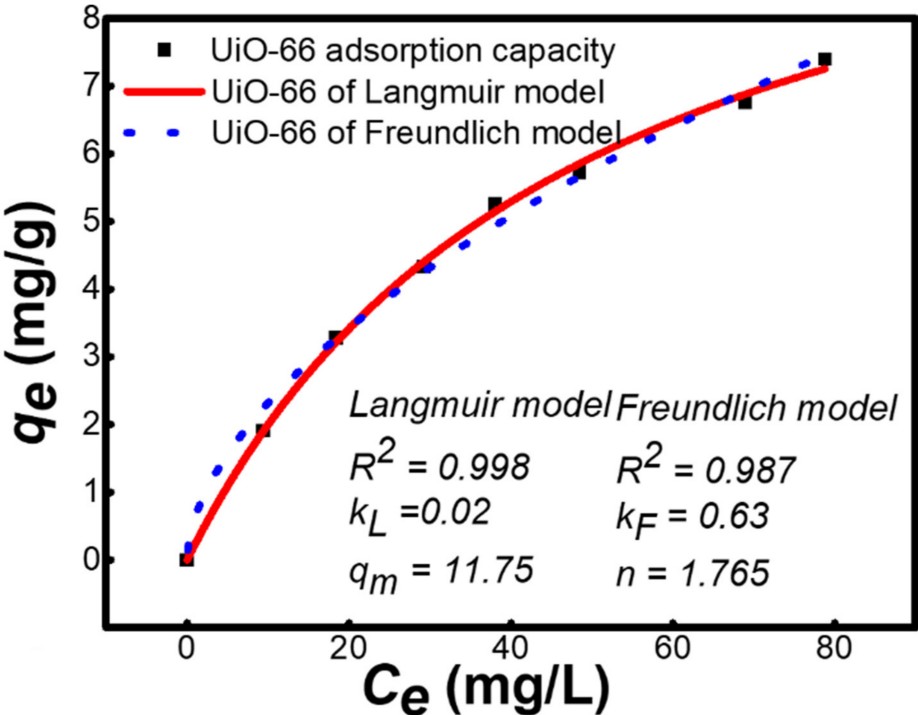

**Figure 4.** Adsorption isotherm of UiO-66 (pH = 3, *t* = 24 h, *m* (UiO-66) = 10 mg, *V* (solution) = 20 mL and *T* = 298.15K).

**Table 2.** Parameters of isotherm models for UiO-66.

| Langmuir Model | | Freundlich Model | |
| --- | --- | --- | --- |
| $R^2$ | 0.996 | $R^2$ | 0.987 |
| $q_e$ (mg·g$^{-1}$) | 11.75 | $K_f$ (mg·g$^{-1}$)·(mg·L$^{-1}$) | 0.63 |
| $K_L$ (L·mg$^{-1}$) | 0.02 | $n$ | 1.765 |

**Table 3.** A summary table of the adsorption capacities of UiO-66 for In(III) and other adsorbents reported in the previous literature.

| Adsorbents | Adsorption Capacity | Reference |
| --- | --- | --- |
| Chitosan-coated bentonite (CCB) | 1–6 mg·g$^{-1}$ | [15] |
| TiO$_2$ | 4566 µg g$^{-1}$ | [16] |
| CoFe$_2$O$_4$−zeolite composite | 94.78 µg g$^{-1}$ | [18] |
| TP207 | 55 mg·g$^{-1}$ | [36] |
| Poly(vinyl phosphonic acid-co-methacrylic acid) | 53.97 mg·g$^{-1}$ | [37] |
| Modified solvent impregnated resins (MSIRs) | 26.25 mg·g$^{-1}$ | [38] |
| Amberlite IRA-400AR | 14.93 mg·g$^{-1}$ | [39] |
| Coated solvent impregnated resins | 23.8 mg·g$^{-1}$ | [13] |
| UiO-66 | 11.75 mg·g$^{-1}$ | This study |

### 3.2.3. Effect of Coexisting Cationic

To evaluate the practical application value of UiO-66 in In(III) adsorption, the influence of interfering ions is an important factor. Therefore, we conducted experiments on the influence of coexisting ions on the UiO-66 adsorption of In(III). As shown in Figure 5, the results indicated that these cations, such as magnesium ions and aluminum ions, did not disturb the adsorption of In(III). This adsorption of UiO-66 toward indium can be explained by the hard and soft acids and bases (HSAB) since indium are hard Lewis acids and UiO-66 contain the hard Lewis base atom O.

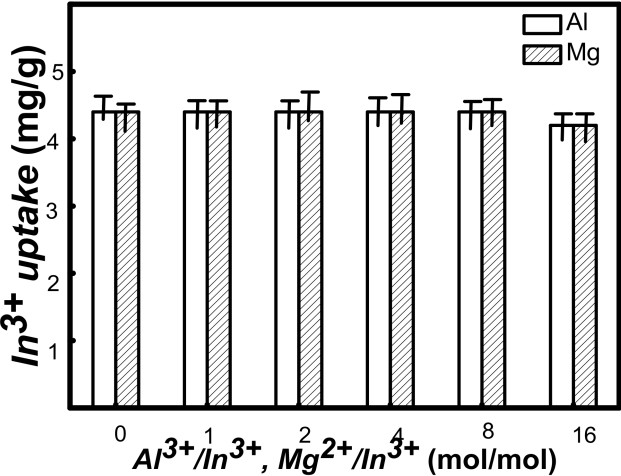

**Figure 5.** Effect of coexisting cationics of $Al^{3+}$ and $Mg^{2+}$ on the adsorption of $In^{3+}$ (pH = 3, $m$ (UiO-66) = 20 mg, $V$ (solution) = 40 mL, $C_0$ = 28 mg·L$^{-1}$ and $T$ = 25 °C).

### 3.2.4. Regeneration Experiments

Figure 6a shows the recycling performance diagram of the UiO-66 adsorbent. The adsorption capacity was unchanged after four cycles of adsorption–desorption, which shows that UiO-66 can be regenerated for the absorption of In(III) perfectly. The regeneration experiment was carried out by separating UiO-66 from an $In^{3+}$ solution of pH 3 by centrifugation, and then by soaking it in an HCl solution of pH 1 for desorption.

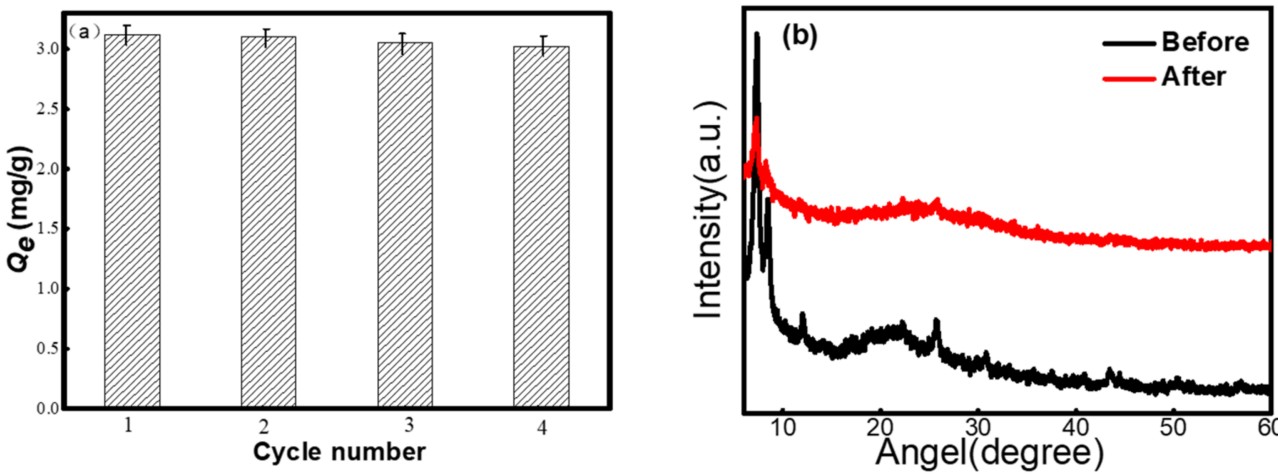

**Figure 6.** (**a**) Desorption and reuse efficiency of UiO-66 ($C_0$ = 20 mg·L$^{-1}$, $V$ = 20 mL, $m$ (UiO-66) = 10 mg, pH = 3 and 298.15 K); (**b**) XRD of UiO-66 before and after In(III) adsorption.

Hydronium ions can compete with adsorbents. The influence of pH value on metal ion absorption is as follows:

$$\text{UiO-66 with -COOH} + \text{In}^{3+} + \text{H}_2\text{O} = \text{UiO-66 with -COO-In}^{2+} + \text{H}_3\text{O}^+ \tag{6}$$

The equilibrium of this reaction is shifted to the left for an acidic (lower pH) solution due to a higher $\text{H}^+$ concentration. Therefore, the In(III) can be desorbed from UiO-66. On the other hand, at higher pHs, there is a decrease in the $\text{H}^+$ ions in the solution, and both the competition for binding sites and the electrostatic repulsion decrease; thus, metal ion adsorption is favorable. The equilibrium of this reaction is shifted to the right again, so UiO-66 can adsorb In(III) again.

Figure 6b shows the XRD of UiO-66 before and after In(III) adsorption. The samples showed typical peaks positioned at $2\theta = 7.4°$, $8.5°$ and $25.7°$, which verified that UiO-66. After In(III) adsorption, the typical peaks positioned at $2\theta = 7.4°$, $8.5°$ and $25.7°$ of UiO-66 remain unchanged, indicating that the adsorption process did not change the crystal structure of UiO-66 [40].

### 3.3. Effect of the Adsorption of UiO-66

The pH value was one of the factors that affected the $\text{In}^{3+}$ adsorption capacity of UiO-66. Indium can be presented as forms of $\text{In}^{3+}$, $\text{In(OH)}^{2+}$, $\text{In(OH)}_2^+$, and $\text{In(OH)}_3$ species at different pH values of solutions. It is worthwhile to mention that indium mainly exists as $\text{In}^{3+}$, $\text{In(OH)}^{2+}$, and $\text{In(OH)}_2^+$ species at pHs below 3. When the pH of a solution > 3, $\text{In(OH)}_3$ precipitation begins to form [15].

In acidic solutions, the uptake of $\text{In}^{3+}$ is low. This may be attributed to the adsorption sites of the UiO-66 surface's being occupied by hydrogen ions, so the adsorption performance under acidic conditions is reduced [41]. Considering the interference of hydrolysis and acidity, pH = 3 was set as the best adsorption condition.

### 3.4. Adsorption Mechanism

To further explore the adsorption mechanism of In(III) on UiO-66-MOFs, FT-IR and XPS were used to test the adsorption mechanism of UiO-66-MOFs. The XPS spectra of UiO-66 before and after the adsorption of In(III) are shown in Figure 7a. The full scan spectrum (Figure 7a) clearly shows that the UiO-66 sample was primarily composed of C and Zr. Concurrently, a new In 3d peak appeared at 452.8 eV, 445.1 eV following adsorption, which indicated that the In(III) was successfully adsorbed onto the surfaces of UiO-66. After XPS-peak-differentiating analysis (Figure 7b), the In 3d spectrum was decomposed to correspond to In(III) (452.8 eV, 445.1 eV) [29,40,42,43]. As shown in Figure 7c, the FT-IR results showed that the C=O absorption band bonded to the In(III) shifted from 1716 cm $^{-1}$ to 1697 cm $^{-1}$ after adsorption [29]. In the FTIR spectra of the pristine UiO-66, the absorption band at 1716 cm $^{-1}$ corresponding to the stretching vibration of C=O in free –COOH groups [44] This indicated that the C=O of free –COOH of UiO-66 was involved in the adsorption of In(III) by cation exchange. Because of the residual DMF solvent, a stretching vibration of amido linkage was observed for UiO-66 before being used for adsorption. And it can disappear after being dried [44]. The UiO-66 after adsorb of indium was put into the oven to dry, so DMF solvent volatilized, thus the peak of 1650 disappeared. The peak at 452.8 eV, 445.1 eV (In $3d_{3/2}$, In $3d_{5/2}$) in the In 3d XPS spectra might also be attributed to $-\text{COO–In}$ complexes.

### 3.5. Adsorption and Desorption of In(III) by UiO-66

Based on the above analysis, a schematic for the adsorption and desorption of In(III) by UiO-66 was illustrated, which is presented in Figure 8. In brief, the UiO-66 possessed free $-\text{COOH}$, which serve as sites for In(III) adsorption in a solution of pH 3. When pH = 1, the solution released a massive amount of protons ($\text{H}^+$) to achieve the desorption of In(III) along with the breaking of $-\text{COO In}$ at 25 °C.

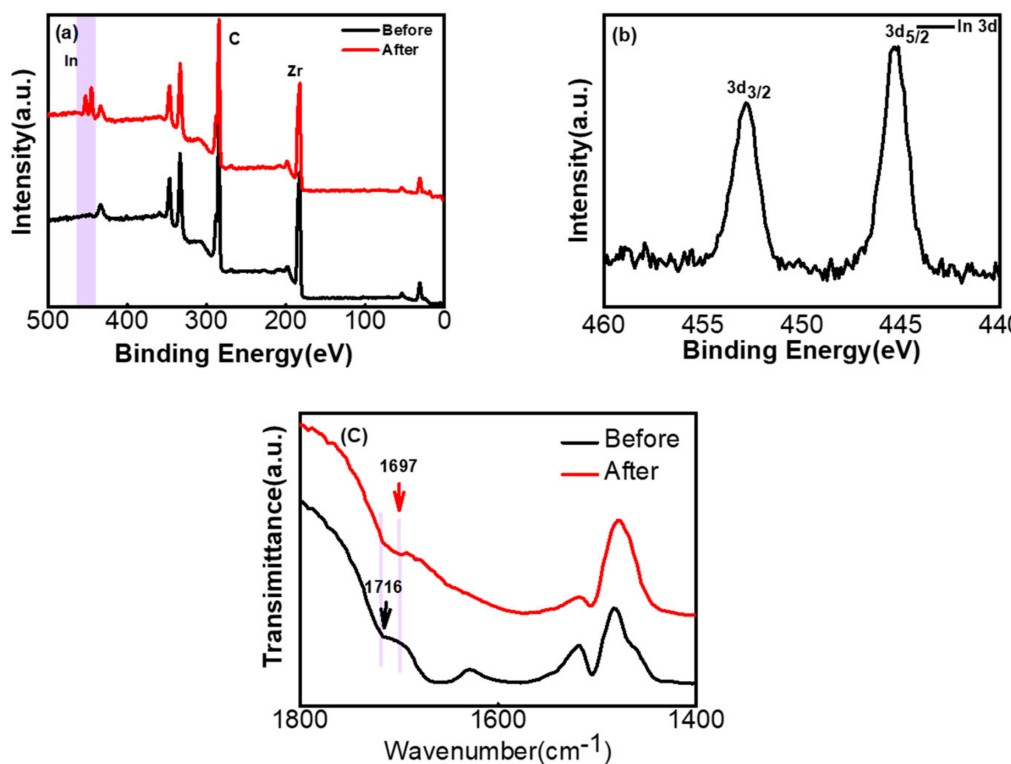

**Figure 7.** Characterization of UiO-66 before and after In(III) adsorption. (**a**) Wide-scan XPS spectra; (**b**) the high resolution In 3d spectrum; (**c**) FTIR spectra.

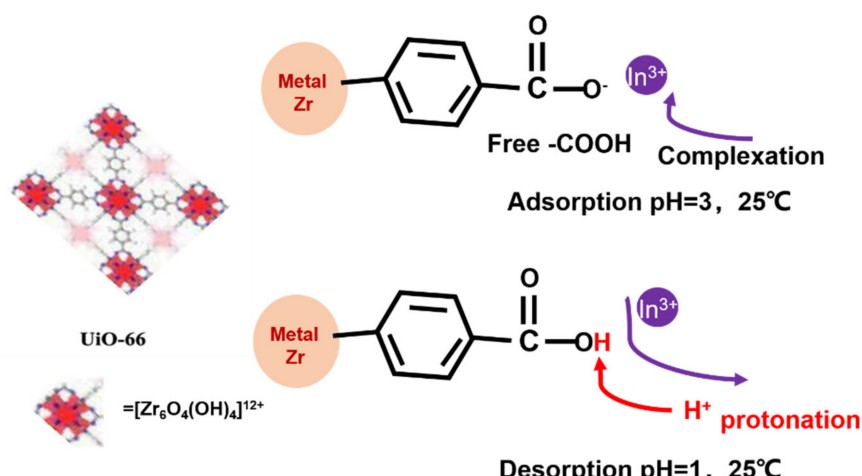

**Figure 8.** Schematic illustration of adsorption and desorption of In(III) by UiO-66.

When the solution of pH increased again, the indium and the free −COOH group of UiO-66 recombined with indium ions as the active site to form a complex.

## 4. Conclusions

In this work, UiO-66 was synthesized by solvothermal synthesis and tested for acid resistance. It was found that the structure of UiO-66 remained intact under acidic conditions, and it was able to adsorb indium in acidic aqueous solutions. Different isotherm models were used to simulate the adsorption equilibrium curve. The greatest adsorption capacity of 11.75 mg·g$^{-1}$ was calculated by the Langmuir adsorption isotherm under the condition of pH = 3. The adsorption kinetics experiment preferably fits the second-order kinetic model. In addition, we found that common coexisting ions, such as magnesium ions

and aluminum ions, did not disturb the adsorption of In(III). Furthermore, UiO-66 can be regenerated easily to absorb In(III) again perfectly. The research conclusions have new guiding significance for the use of UiO-66 as an adsorbent of indium. This research developed a new method for recycling In(III) from LCD electronic waste.

**Author Contributions:** Conceptualization, W.Z.; methodology, W.Z.; software, W.Z.; validation, W.Z., L.X. and C.C.; formal analysis, W.Z.; investigation, W.Z.; resources, W.Z.; data curation, W.Z.; writing—original draft preparation, W.Z.; writing—review and editing, W.Z.; visualization, L.X. and Q.W.; supervision, M.F.; project administration, M.F.; funding acquisition, M.F. All authors have read and agreed to the published version of the manuscript.

**Funding:** This work was financially supported by the Strategic Priority Research Program (A) of the Chinese Academy of Sciences (No. XDA23030302), the Key Programs of the Chinese Academy of Sciences (KFZD-SW-315), and the start-up Foundation from Huaqiao University (20BS109).

**Institutional Review Board Statement:** Not applicable.

**Informed Consent Statement:** Not applicable.

**Data Availability Statement:** Data presented in this article are available at request from the corresponding author.

**Acknowledgments:** This work was supported by technology from the Analysis and Test Center of Institute of Urban Environment, Chinese Academy of Sciences, Xiamen 361021.

**Conflicts of Interest:** The authors declare no conflict of interest.

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
