# Peer review of "Adsorption of Indium(III) Ions from an Acidic Solution by Using UiO-66"

_metals, doi:10.3390/met12040579_

Round 1

Reviewer 1 Report

This study explains the adsorption behavior of UiO-66 for Indium(III) in acidic solutions. The metal recovery using metal-organic frameworks (MOFs) including UiO-66 is a novel approach. This study will be of interest to the readers of this journal. However, modification for several points is required for the publication.

  1. Advantages in the adsorption of MOFs
    The reasons for use of MOFs are not described.
  2. Zr concentration in the acidic solutions after contacting with UiO-66
    The concentration is important data for further supporting the acidic resistance of UiO-66.  
  3. Adsorption equilibrium
    In Figure 3, no plateau by the plots was obtained. We cannot judge whether adsorption equilibrium was achieved or not. 
  4. 3.2.3
    Discussion on adsorption selectivity of UiO-66 for In(III) toward Al(III) and Mg(II) is recommended to be included.
  5. Figure 7
    Precipitation percentages are not adequate because the results change depending on the initial In(III) concentration. This should be displayed with solubility or concentration of In(III). 
  6.  3.3
    The adsorption condition was selected by In species and adsorption efficiencies of UiO-66 for In(III). However, we cannot confirm the efficiencies. The efficiencies against pH need to be displayed.

Reviewer 2 Report

In this work, metal-organic frameworks (MOF) UiO-66 were prepared and used for the separation and adsorption of indium. The results show that UiO-66 can resist acid and keep the structure of UiO-66 unchanged even at strong acidity of pH=1. The adsorption performance of UiO-66 to Indium (III) was evaluated with the maximum adsorption capacity of 11.75 mg·g−1. However, the manuscript contains serious drawbacks concerning the experiments and discussions presented and requires major revisions. Considering the environmental and economic factors for the use of MOFs for metal recovery, the manuscript can be accepted for publication after major revisions. The specific comments are attached below.

  1. The authors are suggested to additionally explain the mechanism of indium metal adsorption over the UiO-66 MOF as no such information is provided in the paper.
  2. It is also suggested to add a schematic representation for the plausible mechanism of removal of indium using UiO-66 MOF.
  3. Many important and relevant citations are missing, following publications can be considered to cite in the revised manuscript.

https://doi.org/10.1016/j.susmat.2021.e00378

https://doi.org/10.1016/j.jhazmat.2021.125941

https://doi.org/10.1016/j.jhazmat.2020.123605

https://doi.org/10.1016/j.apsusc.2020.146974

  1. The manuscript contains serious grammatical errors and typo errors, Line number 137: Characterization of Discussion, 3.2.3. Effect of Coexisting Cationic. The manuscript has to be spell-checked and grammar-checked thoroughly.
  2. The authors need to compare this work with other reported works for the removal of indium in tabular format pointing out the advantages delivered by this work.
  3. The authors need to include error bars in the experiments conducted by reproducing the experiments in triplicates.
  4. 2.4. Regeneration Experiments. The authors have given no information on the procedures used for the regeneration of the adsorption with the possible mechanism of desorption.
  5. The authors are also suggested to additionally characterize the adsorbent after indium adsorption with FTIR spectroscopy to analyze the possible binding of indium ions over the framework.
  6. The authors need to characterize the MOF using PXRD analysis after the adsorption of indium ions to analyze any change in the crystal structure of the material.

Round 2

Reviewer 1 Report

The authors entirely revised the manuscript with the comments of reviewers. However, several mistakes were found. The following issues need to be resolved.

P. 4, L. 187
FR-IR needs to be corrected to FT-IR.

P. 8, L. 260
Hydronium ions mean H3O+. So, you need to change the eq.(6). 
This is better for the Hydrogen ions.

P. 9, L. 279 and 281
The two In(OH)2+ were described. One of those should be In(OH)2+.

P. 9, L. 290
Figure 8a should be Figure7a.

P. 9, L. 292
In3d needs to be corrected to In 3d.

In Figure 8
The deprotonation should be protonation, because H+ bound to the carboxylate.

In Figure 7(c)
A peak in the original UiO-66 is obviously present around 1650 cm-1 . If possible, the peak should be considered, to improve the explanation.

Reviewer 2 Report

The revision has been done to my satisfaction and I feel the manuscript can be accepted for publication. 

Author Response

Thanks!